# A Multi-Token Coordinate Descent Method for Vertical Federated Learning

**Pedro Valdeira**[1,2]**, Yuejie Chi**[1]**, Cláudia Soares**[3]**, João Xavier**[2]

[1]Carnegie Mellon University, [2]Instituto Superior Técnico, [3]NOVA School of Science and Technology
pvaldeira@cmu.edu

## Abstract

Communication efficiency is a major challenge in federated learning. In client-server schemes, the server constitutes a bottleneck, and while decentralized setups spread communications, they do not reduce them. We propose a communication-efficient algorithm for semi-decentralized vertical federate learning dealing with feature-distributed data. Our multi-token method can be seen as a parallel Markov chain (block) coordinate descent algorithm. In this work, we formalize the multi-token semi-decentralized scheme, which subsumes the client-server and decentralized setups, and design a feature-distributed learning algorithm for this setup. Numerical results show the improved communication efficiency of our algorithm.

## 1 Introduction

Federated Learning (FL) is a machine learning paradigm where data is distributed across a set of clients who collaborate to learn a model without sharing local data [23]. Most FL literature considers data distributed by samples (horizontal FL), where each client holds all the features of a subset of the samples, yet recently there has been a growing interest on feature-distributed setups (vertical FL), where each client holds a subset of the features for all samples [1, 5, 7, 13].

FL often deals with the client-server setup with a star-shaped topology. However, such schemes have a single point of failure and suffer from a communication bottleneck on the server [19]. On the other hand, there is extensive literature on decentralized optimization—from earlier work motivated by applications such as wireless sensor networks and multiagent control [9, 25, 28], to recent work motivated by FL [17, 18]. Yet, these algorithms converge slowly in sparse and large networks [26] and, although they spread the communication load across the network, they tend to have a poor communication efficiency [37].

When concerned with the communications between clients, the use of a token method [3, 14, 16, 22, 24, 29], where a model-describing *token* follows a random walk over a communication graph (undergoing local updates), allows for better communication efficiency [14] than the more common consensus-based methods [9, 17, 25, 28], where asymptotic agreement is reached through successive local averaging. Yet, the convergence rate of token methods degrades even faster for larger and sparser networks, due to a lack of parallel communications. Multi-token methods [6, 14, 34] mitigate this problem by running multiple tokens simultaneously and combining them.

Motivated by the above observations, we propose a Semi-Decentralized FL (SDFL) multi-token algorithm for vertical FL. By using both client-server *and* client-client communications, we reduce the communications at the server while mitigating the slow convergence of decentralized algorithms in sparse and large networks. Our algorithm might be of particular interest for applications, for example, using time series data measured by personal devices to learn a model of some "cross-client"

Workshop on Federated Learning: Recent Advances and New Challenges, in Conjunction with NeurIPS 2022 (FL-NeurIPS'22). This workshop does not have official proceedings and this paper is non-archival.

phenomenon of interest (e.g. meteorological). Here, each sample would correspond to the data collected across the devices at a given timestamp.

Our main contributions are as follows.

- We formalize the multi-token semi-decentralized federated learning scheme, which is robust to server failures and flexible in the degree of dependence on the server, recovering both client-server and decentralized FL as particular cases.
- We further design a Multi-Token Coordinate Descent (MTCD) algorithm for vertical FL with improved communication efficiency over state-of-the-art methods.
- Numerical experiments are performed on both synthetic and real data for a variety of communication setups, showing the improved communication efficiency of MTCD as well as the effectiveness of resorting to multiple tokens.

**Related works.** Recently, SDFL approaches have been proposed to lower communication costs and deal with data heterogeneity [11, 20] and to handle intermittent connections, latency, and stragglers [4, 35]. Some SDFL works deal with (multi-layered) hierarchical networks [15, 36].

Coordinate Descent (CD) methods [33], where (blocks of) coordinates are updated sequentially, rather than simultaneously, are natural candidates for optimization in feature-distributed learning. The block selection is most often cyclic [2] or independent and identically distributed random [27, 30]. In contrast, [32] considers block selection following a Markov chain. Several extensions to CD have been proposed, such as acceleration and parallelization [10] and distributed CD methods [6, 21].

## 2 Single-token coordinate descent (STCD)

We now introduce a simple, particular case of our algorithm, closely related to [32] and the application mentioned therein, taken from [22]. Yet, we work in the primal domain and on a feature-distributed setting. In this section, we use the terms client and agent interchangeably.

### 2.1 Problem statement

**Model.** Our goal is to minimize the following regularized generalized linear model:

$$f(\boldsymbol{\theta}) \triangleq \sum_{n=1}^{N} \ell\left(\boldsymbol{x}_n^{\top} \boldsymbol{\theta}\right) + r(\boldsymbol{\theta}), \tag{1}$$

where $\boldsymbol{\theta}$ is a parameter of interest and we have $N$ samples $\boldsymbol{x}_n$, $n \in \{1, \ldots, N\}$. Our loss function $\ell$ is a (possibly nonconvex) smooth and block smooth loss function with a nonempty set of minimizers and $r$ is a separable (possibly nonsmooth) closed proper function, $r(\boldsymbol{\theta}) = \sum_{k=1}^{K} r_k(\boldsymbol{\theta}_k)$, which we assume to have a simple proximal mapping.

**Setup.** In this section, we do not require the existence of a server. The clients $[K] \triangleq \{1, \ldots, K\}$ learn the model in a fully decentralized manner, communicating through channels described by a static, undirected communication graph $\mathcal{G} = (\mathcal{V}, \mathcal{E})$, where $\mathcal{V} = [K]$ is the vertex set and $\mathcal{E}$ the edge set. We assume $\mathcal{G}$ is connected and denote the set of neighbors of agent $k$ by $\mathcal{N}_k \triangleq \{i : \{i, k\} \in \mathcal{E}\}$.

Our dataset $\boldsymbol{X} \in \mathbb{R}^{N \times d}$ with samples $\boldsymbol{x}_n \in \mathbb{R}^d$, $n \in [N]$, is distributed by features across the clients. More precisely, each sample $\boldsymbol{x}_n$ is partitioned as $\boldsymbol{x}_n = (\boldsymbol{x}_{nk})_{k=1}^{K} \in \mathbb{R}^d$, where $\boldsymbol{x}_{nk} \in \mathbb{R}^{d_k}$ and $\sum_{k=1}^{K} d_k = d$. Consequently, we can equivalently define the partitioning as $\boldsymbol{X} = [\boldsymbol{X}_1 \ldots \boldsymbol{X}_K]$, where $\boldsymbol{X}_k \in \mathbb{R}^{N \times d_k}$. Our parameter $\boldsymbol{\theta} \in \mathbb{R}^d$ is partitioned similarly to $\boldsymbol{x}_n$, that is, $\boldsymbol{\theta} = (\boldsymbol{\theta}_k)_{k=1}^{K} \in \mathbb{R}^d$ with $\boldsymbol{\theta}_k \in \mathbb{R}^{d_k}$. The labels $\boldsymbol{y} \in \mathbb{R}^N$ are part of the loss function, which we assume all agents to know.

### 2.2 Algorithm

We now propose a decentralized token method algorithm to minimize (1). Our *token* carries $\boldsymbol{z} \triangleq \boldsymbol{X}\boldsymbol{\theta} \in \mathbb{R}^N$ and performs a random walk over the graph, being updated and communicated by each agent after performing local computations. We denote by $k^t$ the agent holding the token at iteration $t$.

---

**Algorithm 1** Single-token Coordinate Descent (STCD)

---

Initialize $\boldsymbol{\theta}^0 = \mathbf{0}$, $\boldsymbol{z}^0 = \mathbf{0}$, arbitrary $k^0$, and choose step-size $\mu$
**for** $t \geq 0$ **do**
    **if** $t \bmod Q = 0$ **then**
        Agent $k^t$ sends $\boldsymbol{z}^{t+1}$ to agent $k^{t+1} \sim \mathcal{U}(\mathcal{N}_{k_t})$
    **else**
        $k^{t+1} = k^t$
    **end if**
    $\boldsymbol{\theta}_k^{t+1} = \begin{cases} \operatorname{prox}_{\mu r_k}(\boldsymbol{\theta}_k^t - \mu \nabla_k L(\boldsymbol{\theta}^t)) & \text{if } k = k^t \\ \boldsymbol{\theta}_k^t & \text{if } k \neq k^t \end{cases}$      $\{\boldsymbol{z}^t \text{ used when computing } \nabla_{k^t} L(\boldsymbol{\theta}^t)\}$
    $\boldsymbol{z}^{t+1} = \boldsymbol{z}^t + \boldsymbol{X}_{k^t}(\boldsymbol{\theta}_{k^t}^{t+1} - \boldsymbol{\theta}_{k^t}^t)$
**end for**

---

**The token suffices to compute partial gradients.** From the definition of $\boldsymbol{z}$, we have at iteration $t$ that $\boldsymbol{z}^t = \sum_{k=1}^K \boldsymbol{X}_k \boldsymbol{\theta}_k^t$. Letting $L(\boldsymbol{\theta}) \triangleq \sum_{n=1}^N \ell\left(\boldsymbol{x}_n^\top \boldsymbol{\theta}\right)$, we see that, given $\boldsymbol{z}^t$, agent $k$ can compute the gradient of the loss function with respect to (w.r.t.) its local parameter $\boldsymbol{\theta}_k$:

$$\nabla_k L(\boldsymbol{\theta}) \triangleq \nabla_{\boldsymbol{\theta}_k} \sum_{n=1}^N \ell\left(\boldsymbol{x}_n^\top \boldsymbol{\theta}\right) = \sum_{n=1}^N \nabla_z \ell\left(z_n\right) \boldsymbol{x}_{nk}, \qquad z_n = \boldsymbol{x}_n^\top \boldsymbol{\theta},$$

allowing agent $k$ to (locally) perform a proximal CD step w.r.t. $\boldsymbol{\theta}_k$, as long as it also holds $\boldsymbol{\theta}_k^t$:

$$\boldsymbol{\theta}_k^{t+1} = \operatorname{prox}_{\mu r_k}(\boldsymbol{\theta}_k^t - \mu \nabla_k L(\boldsymbol{\theta}^t)), \quad \operatorname{prox}_g(\cdot) \triangleq \operatorname{argmin}_u g(u) + \frac{1}{2}\|u - \cdot\|^2.$$

Given that we only update $\boldsymbol{\theta}_k$ at agent $k$, $k^t$ always knows $\boldsymbol{\theta}_{k^t}^t$.

We now describe STCD, summarized in Algorithm 1, where $\mathcal{U}$ denotes the uniform distribution.

**Initialization.** To start STCD at a given client $k^0$, we need to ensure that $\boldsymbol{z}^0$ which, as we saw, contains all the information needed to do a (proximal) CD step, and $\boldsymbol{\theta}^0$ verify $\boldsymbol{z}^0 = \boldsymbol{X}\boldsymbol{\theta}^0$. To do this without requiring any information from the other clients, we must initialize $\boldsymbol{\theta}^0 = \mathbf{0}$ and $\boldsymbol{z}^0 = \mathbf{0}$.

**Updating and communicating the token.** After performing a local (proximal) coordinate descent step, which gives agent $k^t$ the iterate $\boldsymbol{\theta}_{k^t}^{t+1}$, we need to update the token to ensure that $\boldsymbol{z}^{t+1} = \boldsymbol{X}\boldsymbol{\theta}^{t+1}$. Observing that $\boldsymbol{z}^t$ allows for a local update $\boldsymbol{z}^{t+1} = \boldsymbol{z}^t + \boldsymbol{X}_{k^t}(\boldsymbol{\theta}_{k^t}^{t+1} - \boldsymbol{\theta}_{k^t}^t)$ whose result can then be sent to agent $k^{t+1}$, we see that, by induction, $\boldsymbol{z}^t$ can be kept up-to-date throughout our algorithm. Additionally, this also means that an agent can actually perform $Q$ multiple coordinate descent steps w.r.t. its local parameters, $\boldsymbol{\theta}_k$.

In essence, STCD is a technique allowing for Markov Chain Coordinate Descent [32] to be performed in feature-distributed setups. In terms of the progress made in the parameter space, Algorithm 1 differs from [32] only in that it requires initializing $\boldsymbol{\theta}^0 = \mathbf{0}$ and in that we allow for $Q > 1$.

Let $f$ be a function that is $L_g$-smooth and block $L$-smooth for all blocks $k$, with a nonempty set of minimizers, and let $r$ be a separable closed proper function. It is shown in [32] that, for $Q = 1$, Algorithm 1 converges almost surely to a solution of (1).

### 2.3 Limitations

The decentralized token algorithm in this section has an appealing simplicity. However, while it outperforms state-of-the-art feature-distributed learning algorithms in a variety of setups, as we will see in Section 4, its performance deteriorate faster with network connectivity than these decentralized consensus-based algorithms. Yet, this simple algorithm acts as a stepping stone to the more general multi-token algorithm we now present in Section 3, which mitigates this problem.

## 3 Multi-token coordinate descent (MTCD)

In this section, instead of a fully decentralized setup, we deal with SDFL and generalize the algorithm used in the previous section in a direction allowing us to leverage the existence of a server.

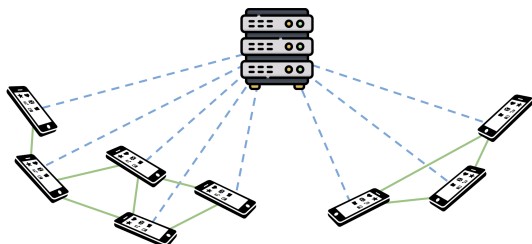

Figure 1: Semi-decentralized setup with $K = 9$. Client-server communications are represented by the dashed blue lines and client-client communications are represented by the unbroken green lines.

### 3.1 Problem statement

In this section, the set of clients and the data and model partitioning are similar to Section 2. However, we now consider a server-enhanced setup (that is, SDFL), illustrated in Figure 1, where we also take into account a central server with links to all the clients. In this section, we do not require the graph describing the communications between the clients to be connected.

Considering the same model (1) as in the previous section, we now develop an algorithm to leverage the existence of this server, exploiting it to mitigate the deterioration of the performance of Algorithm 1 in poorly connected networks.

### 3.2 Algorithm

The algorithm introduced in this section can be seen as having two parts. The roaming part is similar to the procedure in Algorithm 1, except for the fact that we run multiple tokens in parallel, each with an associated model. The novelty lies in the averaging part, where we exploit the access to the server by periodically averaging these tokens there.

More precisely, we use $\Gamma$ tokens $\boldsymbol{z}_\gamma \in \mathbb{R}^N$, $\gamma \in [\Gamma]$, each with an associated model instance $\boldsymbol{\theta}_\gamma$ and $\boldsymbol{z}_\gamma \triangleq \boldsymbol{X}\boldsymbol{\theta}_\gamma \in \mathbb{R}^N$. Similarly to what we had in Algorithm 1, we initialize $\boldsymbol{\theta}_\gamma^0 = \boldsymbol{0}$ and $\boldsymbol{z}_\gamma^0 = \boldsymbol{0}$, $\gamma \in [\Gamma]$, and for $S \times Q$ iterations, each such token undergoes a process similar to the one in Algorithm 1. The difference lies in the fact that now, every $S$ hops of the tokens, between each of which each agent performs $Q$ local steps, the $\Gamma$ tokens are averaged at the server and the $\Gamma$ models are averaged locally, at the clients. Crucially, these averaging operations allow us to preserve the relationship defining $\boldsymbol{Z}_\gamma^t$. (This relationship is also preserved during the roaming part, as explained in the previous section.)

This method, a parallel Markov Chain Coordinate Descent for feature-distributed SDFL setups, is summarized in Algorithm 2, where $k_\gamma^t$ denotes the agent holding token $\gamma$ at time $t$.

**Recovering client-server and decentralized setups.** If no client-server communications are available ($S \to \infty$) our algorithm is reduced to a decentralized one. In this setting, even if at a lower rate, our algorithm will still converge, as long as $\mathcal{G}$ is connected, being reduced to $\Gamma$ simultaneous runs of Algorithm 1. While a decentralized asynchronous token averaging is possible (by exploring the random intersections of multiple tokens are at the same client), this is significantly less effective than server averaging in terms of mitigating the effect of poor connectivity. In contrast, if the set of edges $\mathcal{E}$ is empty and we assign a token per agent, we recover the client-server setting.[1]

**Extension to locally nonlinear models.** We can generalize our model while keeping an additive structure w.r.t. $\{\boldsymbol{X}_k, \boldsymbol{\theta}_k\}$, capturing nonlinearities between $\boldsymbol{X}_k$ and $\boldsymbol{\theta}_k$. Letting $\boldsymbol{h}_k \colon \mathbb{R}^{|\boldsymbol{\theta}_k| \times d_k} \to \mathbb{R}^E$, we would minimize:

$$f(\boldsymbol{\theta}) \triangleq \sum_{n=1}^N \ell \left( \sum_{k=1}^K \boldsymbol{h}_k(\boldsymbol{\theta}_k, \boldsymbol{x}_{nk}) \right) + r(\boldsymbol{\theta}),$$

which includes generalized linear models as the particular case $\boldsymbol{h}_k(\boldsymbol{\theta}_k, \boldsymbol{x}_{nk}) = \langle \boldsymbol{\theta}_k, \boldsymbol{x}_{nk} \rangle$, where $E = 1$ and $|\boldsymbol{\theta}_k| = |\boldsymbol{x}_{nk}| = d_k$. However, this would require additional communications, since

---

[1]To be precise, we would have to change $k_\gamma^{t+1} \sim \mathcal{U}(\mathcal{N}_{k_\gamma^t})$ to $k_\gamma^{t+1} \sim \mathcal{U}(\bar{\mathcal{N}}_{k_\gamma^t})$, where $\bar{\mathcal{N}}_{k_\gamma^t} \triangleq \mathcal{N}_{k_\gamma^t} \cup \{k_\gamma^t\}$.

---

**Algorithm 2** Multi-token Coordinate Descent (MTCD)

---

Initialize $\boldsymbol{\theta}_\gamma^0 = \mathbf{0}$, $\boldsymbol{z}_\gamma^0 = \mathbf{0}$, and arbitrary $k_\gamma^0$, for $\gamma \in [\Gamma]$, and choose step-size $\mu$
**for** $t \geq 0$ **do**
    **if** $t \bmod Q = 0$ and $\lfloor t/Q \rfloor \bmod S = 0$ **then**
        clients send $\boldsymbol{z}_\gamma^t$ to the server
        server computes $\boldsymbol{z}_1^t, \ldots, \boldsymbol{z}_\Gamma^t = \frac{1}{\Gamma} \sum_{\gamma=1}^{\Gamma} \boldsymbol{z}_\gamma^t$
        server sends updated $\boldsymbol{z}_\gamma^t$ to the clients
        **for** $k \in [K]$ **in parallel do**
            $\boldsymbol{\theta}_{1k}, \ldots, \boldsymbol{\theta}_{\Gamma k} = \frac{1}{\Gamma} \sum_{\gamma=1}^{\Gamma} \boldsymbol{\theta}_{\gamma k}$
        **end for**
    **end if**
    **if** $t \bmod Q = 0$ **then**
        **for** $\gamma \in [\Gamma]$ **in parallel do**
            Agent $k_\gamma^t$ sends $\boldsymbol{z}_\gamma^{t+1}$ to agent $k_\gamma^{t+1} \sim \mathcal{U}(\mathcal{N}_{k_\gamma^t})$
        **end for**
    **else**
        $k_\gamma^{t+1} = k_\gamma^t$, $\gamma \in [\Gamma]$
    **end if**
    **for** $\gamma \in [\Gamma]$ **in parallel do**
        $\boldsymbol{\theta}_{\gamma k}^{t+1} = \begin{cases} \mathrm{prox}_{\mu r_k}(\boldsymbol{\theta}_{\gamma k}^t - \mu \nabla_k L(\boldsymbol{\theta}_\gamma^t)) & \text{if } k = k_\gamma^t \\ \boldsymbol{\theta}_{\gamma k}^t & \text{if } k \neq k_\gamma^t \end{cases}$   $\{\boldsymbol{z}_\gamma^t \text{ used when computing } \nabla_{k_\gamma^t} L(\boldsymbol{\theta}_\gamma^t)\}$
        $\boldsymbol{z}_\gamma^{t+1} = \boldsymbol{z}_\gamma^t + \boldsymbol{X}_k(\boldsymbol{\theta}_{\gamma k}^{t+1} - \boldsymbol{\theta}_{\gamma k}^t)$, $k = k_\gamma^t$
    **end for**
**end for**

---

a separate averaging of $\boldsymbol{\theta}^t$ and $\boldsymbol{z}^t$ would no longer preserve their relation, leading to a need to recompute the tokens, instead of simply updating them in an online manner. This would require all clients to communicate with the server.

## 4 Experiments

In this section, we compare STCD to Dual Consensus Proximal Algorithm (DCPA) [1], a state-of-the-art decentralized method used as a baseline, and to MTCD.

**Models and datasets.** We perform ridge regression on a dataset generated following the same process as [1],[2] where the number of samples and the dimensionality are $N = 1000$ and $d = 2000$, respectively. We have the following objective function, with $\alpha = 10$:

$$f(\boldsymbol{\theta}) = \frac{1}{2} \|\boldsymbol{X}\boldsymbol{\theta} - \boldsymbol{y}\|_2^2 + \frac{\alpha}{2} \|\boldsymbol{\theta}\|_2^2.$$

We also perform sparse logistic regression on the Gisette dataset [12], where $N = 6000$ and $d = 5000$. Letting $s(z) \triangleq (1 + e^{-z})^{-1}$, we consider the following objective function, with $\beta = 1$:

$$f(\boldsymbol{\theta}) = -\sum_{n=1}^{N} \left[ y_n \log s(\boldsymbol{x}_n^\top \boldsymbol{\theta}) + (1 - y_n) \log(1 - s(\boldsymbol{x}_n^\top \boldsymbol{\theta})) \right] + \beta \|\boldsymbol{\theta}\|_1, \qquad y_n \in \{0, 1\}.$$

**Metrics.** We use CVXPY [8] to obtain $f^\star$, and then compute the suboptimality gap $f(\boldsymbol{\theta}^t) - f^\star$ throughout our experiments. In the previous sections, we considered each local CD step as an STCD iteration, to avoid the visual clutter of having a double counter (for hops and local CD updates). In this section, to allow for a fairer comparison with DCPA, where only communication rounds count as a round (despite the fact that we need to compute a proximal operator with a gradient descent

---

[2]Each entry of $\boldsymbol{X}$ is drawn uniformly at random from $\{0, 1\}$ and each entry of $\boldsymbol{y}$ is drawn from a standard normal distribution (in both cases the entries are independent).

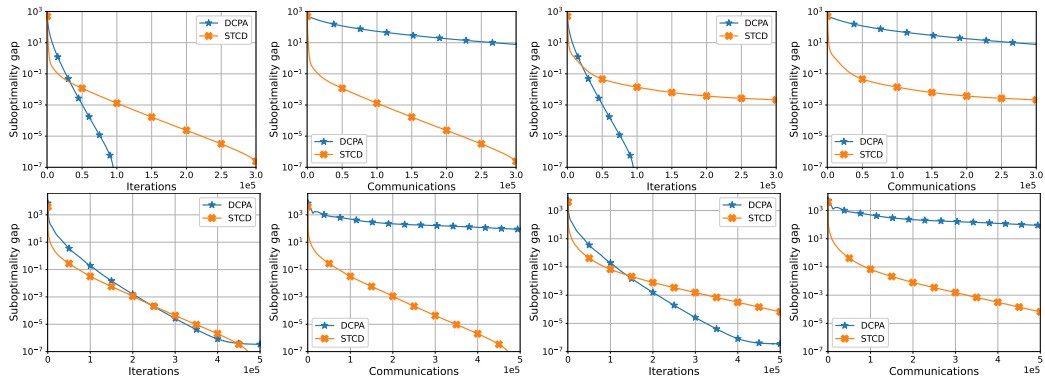

Figure 2: Suboptimality with respect to iterations and communications for a random network with $p = 0.4$ (four figures on the left) and a chain network (four figures on the right), both with $K = 20$. The top row concerns ridge regression and the bottom row concerns sparse logistic regression. Each communication is size $N$: STCD has one communication per iteration and DCPA has $2|\mathcal{E}|$.

subroutine), we count as an STCD iteration each hop in the network. We use the same definition of iteration for MTCD.

**Comparison with state-of-the-art.** For ridge regression, we randomly generate a graph as in [31], with connectivity ratio $p = 0.4$, and a chain graph, both with $K = 20$ nodes and $d_k = 100$ for all $k$. For STCD, we use $\mu = 10^{-5}$ and $Q = 20$. For DCPA, we use $\mu_w = 0.01, \mu_y = 0.0003, \mu_x = 0.03$.

For sparse logistic regression, we similarly generate a random graph and chain graph, both with $K = 20$ nodes (now $d_k = 250$ for all $k$). For STCD, we use $\mu = 10^{-4}$ and $Q = 30$. For DCPA, we use $\mu_w = 0.001, \mu_y = 0.00003, \mu_x = 0.003$.

In Figure 2, we see that, while STCD does not improve upon DCPA in terms of progress per iteration, it significantly outperforms it in terms of communication efficiency. Yet, we can also see that STCD is particularly vulnerable to poorly connected networks, as evidenced by a performance deterioration when going from a random network to a chain network.

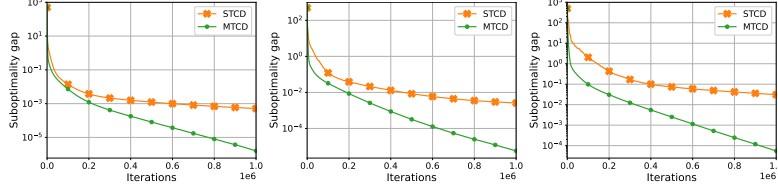

Figure 3: Suboptimality with respect to iterations for $K = 20, 40, 80$ chain networks ($K$ increasing from left to right). All three plots concern the ridge regression model.

**Comparing STCD with MTCD.** We again deal with the ridge regression problem above and resort to the same dataset. However, we now focus on chain graphs (for MTCD, complemented by client-server communications), as they are poorly connected setups where the drawback of STCD is evident. We use $\mu = 10^{-5}$ and $Q = 20$ for both STCD and MTCD and use $\Gamma = 10, S = 10$ for the latter throughout all three experiments, varying only the number of nodes $K \in \{20, 40, 80\}$.

In Figure 3, we see that, while both algorithms see a drop in performance for graphs with a lower connectivity, MTCD mitigates this effect, consistently outperforming STCD.

## 5   Conclusions

We formalize the multi-token SDFL scheme and propose a communication-efficient SDFL algorithm for feature-distributed data. Numerical results show the improved communication efficiency of our algorithm as well as the power of endowing decentralized methods with periodical client-server communications. A natural extension to this work is a detailed study of how the number of tokens and the frequency of their averaging influence the convergence rate, both empirically and analytically.

## Acknowledgments and Disclosure of Funding

This work is supported in part by the Fundação para a Ciência e a Tecnologia through the Carnegie Mellon Portugal Program; by the grant U.S. National Science Foundation CCF-2007911; and by the CMU-Portugal project CMU/TIC/0016/2021.

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
