# OpenReview forum: "A Multi-Token Coordinate Descent Method for Vertical Federated Learning"
_NeurIPS.cc/2022/Workshop/Federated_Learning — FL-NeurIPS 2022 Poster_

### Official Review · Reviewer_zyMY · 2022-10-16
**Review for "A Multi-Token Coordinate Descent Method for Vertical Federated Learning"**

The paper approaches a somewhat unusual FL setting: vertical, decentralized, valid for linear models only, and with learning done sequentially and not in parallel (for "single-token", at least). It is, thus, a rather limited setting, and no practical examples of where this setting could be useful are provided. There are also no concerns on whether the data being sent from client to client is leaking anything sensitive, which is usually one of the main things we worry about in FL.

It is also a nicely written paper. All ideas are neatly explained and easy to understand. In particular, the idea of averaging the z's coming from different cliques is interesting, albeit not very well motivated. Experiments clearly show what are the advantages and disadvantages of the presented algorithms.

My main concerns are, indeed, how useful are these algorithms in practice, given the limited setting, and privacy leakages not being considered. The rating was given taking all that into consideration.

Minor points:
- FD in line 96 is not defined anywhere
- Why isn't the single token algorithm considered of the main contributions? Is it because it has already been used in the distributed setting (add citation in that case), or because the adaptation is trivial?

---

### Official Review · Reviewer_TBHr · 2022-10-17
**Adaptation of Markov chain coordinate descent to Federated Learning**

In this paper the authors adapt Markov Chain Coordinate Descent to Vertical Federated Learning. The first adaptation is allowing for more than one local step at a given node; the second adaptation is a semi-decentralized approach with multiple models in parallel, where a server exists to enable model averaging. The second adaptation forms a general algorithm that subsumes client-server and decentralized schemes.

The paper is concise, and novelty is properly assessed in relation to previous works. The extension and implications to non-linear cases is however not satisfactorily discussed. Experiments are also not fully developed as we do not get a real grasp on the trade-off between 'communication efficiency' and 'progress per iteration'. The authors could have explored more the STCD trade-off (in relation to DCPA) since it seems to be a key part of assimilating the paper as a relevant and worthwhile method do use/discuss. Also, it is not very surprising that MTCD would outperform STCD, with its extra communication and multiple model averaging at the server. Again, what is interesting is the trade-off between communication and progress.

I believe this would be marginally above a threshold for a workshop paper that aims at 'starting a discussion' over a given method/idea. However, I believe it deserves a better analytical and/or empirical development of the key trade-off (as mentioned above).

---

### Official Review · Reviewer_SjDT · 2022-10-18
**An interesting idea which requires theoretical guarantees**

The authors propose a multi-token semi-decentralized scheme for performing vertical federated learning. Multiple tokens perform a random walk (in parallel) over a general connected graph, with features split between client. For sparse, large networks, one token is not enough, so the authors propose the multi-token paradigm to alleviate this issue, and the use of a coordinating server is to resolve the slow convergence pace associated with fully-decentralized networks.

Pros:
-Vertical federated learning (VFL) is still relatively unexplored. It is very much a fertile ground for research.
-Great presentation. Introduction, motivation, and algorithm are clearly explained.

Cons:
-The contribution is not novel enough, $\textbf{especially}$ without a theoretical guarantee. Convergence for a single token performing a random walk is known. Parallelizing an existing framework (in this case, a single token walk) does not rise to the level of a high level research contribution associated with a conference such as NeurIPS unless one can show, analytically, how it affects model convergence, especially with regards to the number of clients, tokens, etc.
-Continuing off the first point, look to peer works: VAFL and Compressed-VFL both provide convergence guarantees because the setup with VFL is highly unusual in general -- we're splitting feature spaces after all. It's both counterintuitive yet exciting and useful to see such protocols convergence.

This is a neat preliminary idea, though my opinion is that it will not interest others in this field unless there's some guarantee the suggested algorithm will converge. If the idea itself was entirely novel, empirical evidence would be enough, but it's a (mostly) simple modification of an existing approach.

---

### Decision · Program_Chairs · 2022-10-20

Accept (Poster)